# A Qualitative Exploration in Causes of Water Insecurity Experiences, and Gender and Nutritional Consequences in South-Punjab, Pakistan

**DOI:** 10.3390/ijerph182312534

**Published:** 2021-11-28

**Authors:** Farooq Ahmed, Muhammad Shahid, Yang Cao, Madeeha Gohar Qureshi, Sidra Zia, Saireen Fatima, Jing Guo

**Affiliations:** 1Department of Anthropology, Quaid-I-Azam University, Islamabad 44000, Pakistan; jam007@uw.edu; 2Department of Anthropology, University of Washington, Seattle, WA 98105, USA; 3School of Insurance and Economics, University of International Business and Economics (UIBE), Beijing 100029, China; DE202159006@uibe.edu.cn; 4Pakistan Institute of Development Economics, Islamabad 44000, Pakistan; madeeha.qureshi@pide.org.pk; 5Independent Researcher, Islamabad 44000, Pakistan; sidrazia2010@hotmail.com; 6Fazaia Medical College, Air University, Islamabad 44000, Pakistan; 16021@students.au.edu.pk; 7Department of Health Policy and Management, School of Public Health, Peking University, Beijing 100091, China; jing624218@bjmu.edu.cn

**Keywords:** water insecurity, experiences, maternal-child, health, malnutrition, qualitative research, South-Punjab, Pakistan

## Abstract

Although water insecurity has been discussed in general, its impacts on mothers’ physical and mental health, and infants’ and young children’s feeding (IYCF), has largely been ignored. This study explores household water insecurity experiences and their association with optimal health and nutrition of women and children in the Rajanpur district of Punjab Province. Using focus group discussions (FGDs) and key informants (KIIs) interviews from an area with high maternal-child malnutrition prevalence, a qualitative study was conducted to describe local experiences of water acquirement and arrangement, and of the consequences of water insecurity. The findings highlight that rural Western marginalized populations of the Rajanpur district rely on brackish, canal, or flood surface water as the water supply is absent, which intensifies mothers’ work burden and stress, and often makes them victims of violence, stigma, and sickness. Water fetching impacts women in unforeseen ways, impacting the psychosocial and physical health of mothers engaged in maternal breastfeeding. Water insecurity, originally rooted in regional disparities, compounds with gender inequities, which leads to maternal stress and child sickness. Justice in water resources is imperative and urgent in the deprived South of Punjab province for improving public health nutrition.

## 1. Introduction

Food insecurity, maternal poor health, suboptimal water, sanitation, and hygiene (WASH) and IYCF practices, which are considered to be important underlying determinants of malnutrition, have a close association with water insecurity. In order to tackle poverty, ill health, inequality, and environmental degradation, the Sustainable Development Goals (SDGs) were established in 2012 at the United Nations Conference on Sustainable Development in Brazil. SDG-6 ensures “availability and sustainable management of water and sanitation for all” [1]. However, almost 1.8 billion people drink water from contaminated sources worldwide [2]. Access to safe water is fundamental to health, nutrition, and development in low-and-middle-income countries [LMIC] [2]. The availability of safe drinking water is not possible without proper community development interventions, but in low-and-middle-income countries, the peripheries are deliberately ignored in terms of human development. Households and communities need a well-laid-out sanitation system to prevent infections. The macro-level situation of water in Pakistan has implications, especially for deprived districts of Southern Punjab. The poor water management shows a lack of vision and strong will to good governance and development.

Stunting rates in the Rajanpur district (48%) of Pakistani Punjab are the highest [3,4]. The water situation has serious implications for the health of both women and children. The highest stunting prevalence in the Rajanpur district might be strongly associated with a severe water insecurity situation. A previous study in Southern Punjab shows that inadequate water quantity, mainly because of distance, was a robust predictor of height-for-age (HAZ) and the prevalence of diarrheal infections [5].

Good water governance has been discussed as a critical determinant of household water security [6,7,8,9,10]. The first 1000 days of a newborn’s life, known as the window of opportunity, has been observed and taken up as the most crucial time of a child’s growth. The main reason behind half of global child mortality in the year 2011 was inadequate or inappropriate feeding practices [11]. Water might be one of the most significant influencers that can affect optimal IYCF practices in severely water-insecure areas. A study in Bangladesh revealed that food became contaminated with the pathogen *E. coli* [12]. Improper hygiene practices in women and their children greatly affected the well-being of mothers [13,14]. Moreover, the evidence suggests that arsenic contamination of drinking water is a possibility that may result in contaminated breastmilk [15].

Evidence shows that, worldwide, four billion (2/3) people face moderate to severe water scarcity during at least one month in a year. However, in Pakistan, 120 million (>3/4 in the Indus basin) people face severe water scarcity for at least 4 to 6 months per year [16]. Severe water scarcity hits women and children on critical days of pregnancy and lactation. Household water insecurity not only reduces the care capabilities that are required for optimal infant feeding [17], but also co-occurs with food security [10,18,19,20]. It establishes a syndemic relationship [20] and causes inadequate nutritional, developmental, physical, and mental health outcomes [10,14,19,21,22], especially hypertension [18], depression, and anxiety in poor communities [18].

Although the literature on water security at the macro-level is available, anthropological and qualitative accounts, as well as the lived experiences of water insecurity, have not been captured from poor, deprived, and malnourished southern communities in Pakistan. Besides, it is common to associate malnutrition with food insecurity, but maternal factors and infant and young child feeding have never been linked with the bad water situation. Water insecurity does not impact WASH only but also IYCF practices. Against this backdrop, the current study explores the households’ experiences of poor water conditions in general and the impact of water fetching and borrowing on maternal stress and infant feeding in particular.

Water insecurity was shown to impact maternal and child health across four pathways in the Kenyan context: physical and psycho-social health, nutrition, and economic wellbeing [23]. This research aimed to investigate household experiences regarding water’s relationship to optimal maternal-child health and nutrition. We argued that water insecurity is a syndemic and cross-cutting issue, associated with maternal-child malnutrition through multiple pathways: gender inequity, maternal stress, sub-optimal childcare, poor IYCF and WASH practices, and food insecurity.

### Study Setting

Rajanpur, one of the least developed districts of Punjab province in Pakistan, has a high prevalence of maternal-child malnutrition (~50% maternal anemia and child stunting) as well as a high poverty level, where 60% of the total population has been identified as poor [3,24]. The level of human development in the district is also the lowest among all 36 districts of Punjab Province [3]. The latest District Census Report of Rajanpur [25] indicates that the total area of the district is 12,318 square kilometers. The total population of the district is above 2 million, comprising 85 percent rural population and 14 percent urban population. There are 262,490 households in the district. The total number of housing units in the district amounts to 151,733, including only 17 percent that are made of concrete. Forty percent (40%) of houses have electricity. However, only nine percent of houses have piped water facilities and 0.63 percent of houses use gas for cooking [25]. Overall, Rajanpur presents a picture of south Punjab as having insufficient health and education facilities and an increased poverty rate. There are 33 Basic Health Units (BHU), 6 Rural Health Centers, and only one hospital with 40 beds. Even though there are 980 primary schools, the number of elementary and secondary schools are 140 and 59, respectively. The literacy rate is significantly low at 29 percent for males and only 11 percent for females [3].

Farmers face an exceptionally high shortage of water because of a flawed mechanism of water distribution and acquire only 9000 cusecs (against a total capacity of 24,000) of drinking and irrigation water from three canals [26]. The study’s locale has a long mountain range on its West, and the great Indus River flowing parallel in the East, facing chronic environmental disasters in the form of torrential floods from two directions. Parallel to the main Indus Highway and near to Suleiman Mountains, the remote Western areas of the district are geographically secluded. After the monsoon season between July and August, the floods from the Suleiman Mountains make the situation more vulnerable by destroying healthcare facilities, educational institutions, roads, and other facilities. The underground water in western parts of the district is brackish; hence, inhabitants use rain and canal water as the primary source of drinking water. Canal water is not provided all the year but only on a six-monthly basis; therefore, the western parts of the district have to experience moderate to severe water insecurity. Rajanpur has an almost desert-like climate, where the average temperature is 26.0 °C and the precipitation is 205 mm [27] (rainfall is almost negligible in the region).

## 2. Materials and Methods

### 2.1. Data Collection

In total, three FGDs and five KIIs were conducted from April 2018 to May 2018. We used the purposive sampling technique to conduct FGDs. Only the western villages of the Rajanpur district were selected because they were considered the most water insecure by the local mapping. Out of three FGDs, one was conducted with community females, one with community males, and one with Lady Health Workers (LHWs). In each FGD that lasted for two to three hours, no more than ten persons were allowed to participate (Table 1). After a detailed review of the literature on household water insecurity experiences, a questionnaire was designed to collect qualitative data through a probing and discussion methodology. The first FGD was conducted in the most Western water-poor areas, where ten males participated and discussed the vulnerable situation related to water, food, floods, development, and malnutrition. The second FGD was conducted with female members of water-poor communities was assisted by female moderators and issues such as impacts of water insecurity on maternal stress and breastfeeding behaviors were discussed. The third FGD was with LHWs who provided critical and useful information regarding child and mother health and nutrition. During the discussion, they provided data on multiple aspects of breastfeeding, water, sanitation, and hygiene. At the end of each FGD, the participants were thanked and served with tea and a local rice dish.

Using snowball sampling, five key informants (3 females and 2 males) were recruited who were local but from different professions (Table 1). All of the key informants helped in providing important and relevant data about the area, beliefs, practices, and experiences of water insecurity and its causes and consequences. The first author was assisted by a female research assistant, who collaborated and contributed to this fieldwork in a great many ways. The Seraiki language was used to write down the notes, interviews, and discussions. Audio recorders were not used due to the considerations of the sensitivity and comfort of the study’s respondents. Interviews and field notes were immediately translated verbatim into the English language by the principal author and were counter-checked to validate the authenticity of the data. In addition, a debriefing session was held soon after each interview and focus group discussion so that the research team might review the process of the interview, evaluate the key findings, and mitigate discrepancies.

### 2.2. Data Analyses

After an intensive review of all translations and field notes by the first authors (F.A.), the qualitative data were analyzed through a manual thematic analysis approach. Using an inductive method, two co-authors (F.A., S.Z.) assembled various concepts, themes, and interpretations after a detailed perusal of the raw data. Through cross-verification of the narratives and field observations, discrepancies and similar codes were removed with the mutual consensus of the co-authors. The authors analyzed the data and shared their understanding while labeling the meaning units in the data with descriptive codes and conceptual categories. After reaching consensus, the conceptual categories were described, and a reverse check was conducted, in which categories and corresponding significant statements in the data were examined. Field notes were used to draw contextual inferences and interpret the data, ensuring descriptive validation of the study, with significant statements being used as evidence. The analysis was conducted using the inductive approach and different themes emerged from the data until the point at which we realized that new themes could no longer emerge. We analyzed and interpreted the data manually and organized them into different themes and sub-themes.

The objective of this study was to explore the potential causes of water insecurity experiences in the Rajanpur district of South Punjab and how they contributed to suboptimal maternal-child health and nutritional consequences. Therefore, employing a combination of both FGDs and KIIs (a robust methodological approach) in the current study enabled us to obtain more detailed, relevant, and qualitative experiences and extract relevant themes.

### 2.3. Ethical Considerations

This study obtained ethical and IRB approval from the Advance Study and Research Board (ASRB) of Quaid-i-Azam University, Islamabad, Pakistan in its 307th meeting held on October 20, 2016. All respondents were informed about the purpose of the research before participating in the study, and then their oral consent was obtained. Keeping the ethics of the study in view, the privacy, anonymity, and confidentiality of all participants were promised and strictly ensured.

## 3. Results

The final list of participants included 35 respondents (*n* = 35). Excluding 2 KIs and 10 LHWs, the majority of the community participants were illiterate or attended school for only a few years, and were of lower socioeconomic status (income < USD 100/month). They were working either as agricultural laborers or domestic household servants (Table 1). All the interviews were conducted in the local language (i.e., Seraiki). The study developed an understanding of the research setting, community perceptions, and major themes related to the causes and consequences of water insecurity: access and availability of water, locals’ coping strategies, pathways through which water situation affects maternal-child health, and implications for infant feeding practices.

### 3.1. Water Governance and Coping Strategies

#### 3.1.1. Lack of Water Supply and Low Water Quality

The majority of households from western areas of the district had no water supply system. They had no option except to use unsafe and muddy canal or flood water that looked, tasted, and smelled awful. The surface water that they used was unsuitable for drinking purposes. One middle-aged woman reported:

*“We rely on rain or flood water, collected in ditches or canal water. As underground bore water is salty, there is no choice but to use dirty and muddy canal water. Though sweet underground water is available at a few places, however, bore is very expensive. Canal water is polluted and tastes like clay. The hard water is not suitable for feeding and bathing; also, peas and meat are hard to be cooked. We sometimes use underground water when the canal water is not enough.”* (FGD, Female, 50)

Unable to bear the expenses required for boring because the water was very deep (200–300 feet), the poor respondents expressed their miserable conditions and demanded urgent measures to solve this critical situation.

#### 3.1.2. Treatment of Water Is Difficult and Expensive

The use of dirty cans to fetch water was common. The animals and humans drink water from the same source. The bad water condition damaged good health and well-being practices.

*“Frogs and lizards are often found in our water ponds, but we still use this type of water. We know boiling the water is good, but boiling is difficult, expensive, and time-consuming. It requires money and wood to boil water; therefore, we cannot afford this.”* (FGD, Male, 43)

*“My child has been suffering from severe malnutrition because bad water did not suit him. We used bottled water, but it was costly. We need a free water filter plant for drinking purposes as we’ve to pay an inflated amount for fetching water.”* (FGD, Female, 18)

Brackish water caused digestive problems and proved fatal for the infant’s stomach. A mother lamented:

*“Doctors also restricted us to give canal water. We used to boil water in the past, but it is difficult to boil always, so we don’t treat water now. Infants sometimes feel pain in their bellies due to hard water.”* (FGD, Female, 29)

#### 3.1.3. The Politics of Water Injustice

Lack of political will was described as the cause of the water problem. According to locals, no water supply scheme could be mounted because of the indifference, selfishness, and corruption of politicians. One key informant stated:

*“Our politicians are disloyal and insincere to solving this problem. The government should take responsibility for the water supply for this area. The district and provincial government and their inept administration have turned deaf ears to this essential demand. We need a clean water supply here. If we were opulent, we would have left this area. The canal flows only a few months, water stored in the pool becomes very murky over some time and undrinkable.”* (KII, Male, 35)

It was reported that canals in South Punjab run for less than half a year. Water distribution emerged as the most prominent political question. Water was never subject to such competition until the cultivation of cash crops. The leading cause of the water problem was water injustice (Figure 1). Water and Power are strongly connected, as illustrated by a KI:

*“The width of the canal is narrow, and then water is stolen by the strong peoples through making Moga (cuts), often in cooperation with the irrigation department. People bribe them… And as a result, the water level reduces at the tail end… And is insufficient for agriculture. The most disadvantaged group is small peasants, who also often protest against this injustice. It has become a day-to-day practice now and is happening for decades. It impacts our whole life, affecting our food, income, and health.”* (KII, Male, 57)

#### 3.1.4. Coping Strategies: Displacement as Last Resort

The dry season includes the months of May, June, September, October, November, January, and February. The canal water is available only in June and July, but not in January, February, September, and October. People considered it punishing that children, animals, and older people were staying in a place where water was not available. Consequently, they had to move to other places where the water condition was better, even though they became exhausted during long hours of travel and, as a result, could be exposed to diseases. One man narrated his tale during a group discussion:

*“We live here because of our tribe; otherwise, the water situation is not right here, it’s a dry place… We are forced to drink unsafe muddy water collected through rain or flood. We wait only for the situation to get better before seriously undertaking the movement plan. We sit silent, wait, and be patient, and pray for rain. Migration begins when water ends. During this arduous journey, we become homeless, often without food and water, and a toilet. On a long journey, some animals even die due to water scarcity. The ground is our room, and the sky is our roof… Where there is water, there is God. For us, water is everything.”* (FGD, Male, 53)

### 3.2. Water Borrowing and Fetching Cause Gender Vulnerability

#### 3.2.1. Price for Water Borrowing—Stigmas and Difficulties

Poor people felt stigmatized as a result of the rude and disgusting behaviors of other people that were expressed as a result of borrowing water. They complained about how rich people made a mockery of them whenever they requested to fill their cans. They revealed that they sometimes had to listen to abusive words, which pinched them several days after obtaining water. More importantly, females complained that they felt highly insulted when they entered others’ houses for water. Girls were reportedly told that they could receive water free of charge if they were willing to engage in illicit relationships.

These women, who were dependent on better-off people in the village, were used for different purposes such as cleaning courtyards, buying items from the market, and dropping children off at school. They were often forced to provide labor in compensation for being given water.

*“People who give water demand something in return. In return, the borrowing family usually provides shawls with some embroidery work as well as a shirt for males and infants as an act of reciprocity. It is hand-made work that takes time and energy. Some have to make tandoori roti (wheat flour’s loaf) in return for borrowed water, which is a very hectic job on hot summer days, but it is water scarcity that forces us to do so.”* (KII, Female, 53)

#### 3.2.2. Women Face Harassment

Problems with water significantly increased difficulties within and outside households. Children, especially girls, complained to their parents that when they go to fetch water, older boys harass them, and they feel insecure, so elderly members of their households must go along with them to fetch water. One woman reported:

*“Children face harassment while fetching water, and there are chances of harassment and even rape. Women also take a bath when they go to fetch water, which most often causes embarrassment and even fights as rival women protest on water wastage. They felt danger, especially after sunset, to carry water because of male harassment. Husbands often feel not only angry at wives but also cast suspicion when they are late from bringing water. Females, therefore, usually go to fetch water in groups in daylight and avoid facing strangers. The group formation is a safe method for women, so they go to the pool to wash clothes jointly where they feel safe.”* (FGD, Female, 27)

*“Children defecate openly in the house premises….the practice of open defecation is common here. Females usually go far away from their courtyards. They have to halt defecation until they all go together for defecation in a group to a distant place reserved for this activity and return when everyone has finished because sometimes alien males of other villages and ethnicity followed and harassed adolescent girls and women, which sometimes result in the feud and even honor killing (Kala-kali).”* (FGD, Female, 32)

#### 3.2.3. Difficulties in Fetching Water

Although the responsibility for providing water in the house was shared among parents and children, water fetching was very tough and burdensome for women, children, and elderly persons because of the need to make so many trips on a daily basis. One mother stated that it takes a minimum of half an hour to fetch one pot of water, and the average distance they cover each day is two kilometers:

*“Water fetching is a very hectic job, my husband works in another city, and my kids are still small who can’t be left alone at home. Even the father-in-law at his old age helps to fetch water from the bore or pool. Also, bringing water is difficult, especially for old persons, children, and women owing to the distance, and the number of trips to water sites. It requires more than half an hour to return home after traveling two miles.”* (FGD, Female, 24)

#### 3.2.4. Fights and Injuries over Water Fetching

On several occasions, quarrels on the issue of water erupted. One woman recalled she had a severe fight with neighbors on water fetching as it was her turn, but someone tried to take her number. Water fetching causes fatigue and leads to critical health problems, including injuries. Sometimes, trivial fights convert into a family scuffle and feud. Women reported that they often abuse each other during fetching water, and their children also fight with each other and sometimes become injured. The main reason for the clashes is that most women want to save their precious time, as they grow tired of waiting in the queue for the water. They start labeling, abusing, and pushing each other to reach the water first. The scarcity of water and the effort involved in fetching it involve conflicts within this community and caste-based discrimination is observable against women of the lower caste during the fetching of water.

### 3.3. Water Insecurity Impacts WASH and IYCF

#### 3.3.1. Feeding and Hygiene Require Adequate and Safe Water

Bad water situations affected water, sanitation, and hygiene (WASH). Open defecation is a result of a lack of water and proper toilets. Consequently, body purging (*Gheesi*) is performed after open defecation with either mud or a pebble, and this leads to problems concerning feeding. Hygiene and the prevention of infections require water. One health worker said:

*“If water isn’t sufficient for drinking, how they can wash hands…so what benefit is of teaching handwashing lessons to them. A common man is hardly surviving there, how come you advise him to live a better life. Diarrhea, pneumonia, and small intestine problems are common as water is muddy. Besides this, hepatitis is the biggest infection due to bad water. Hygienic methods are not followed.”* (FGD, female, 43)

A sufficient amount of water is needed to maintain hygiene. It was reported that the canal water channeled through the Indus River was muddy, the underground water was brackish, and bottled water was costly. One respondent reported: “the infant is going to the hospital every week because of bad water. We only use pure bottled water when the infant becomes sick.” Feeding practices were not safe; mothers did not take care of the hygiene of baby bottles, plates, as well as glasses for drinking and eating.

*“Water vessels are filthy in these areas. There are a lot of flies around water containers. Hand washing is overlooked due to security and scarcity of water, and that has often made children suffer from sickness.”* (KII, Female, 31)

*“Dirty water caused vomiting and stomach issues. Food prepared in bad water becomes sometimes contaminated... When dry milk was mixed with bad water, infants faced constipation with formula milk, utensils remained unwashed, and children were fed with dirty hands.”* (FGD, Female, 26)

#### 3.3.2. Water Fetching Influences Breastfeeding

The number of trips and time taken to fetch water made mothers uneasy and unhappy during special days and periods of sickness. The time required for water fetching made them stressed, lethargic, and slow, especially during sickness and menstruation. Due to distant travel, they became tired and become sick, and their immunity became poor. The women involved in labor work could pay no proper attention to their children, and fetching water often compromised children’s care, and dehydration was common.

*“Once children had diarrhea, but there was no water in the household. We had to travel, ignoring care and cure. Water scarcity weakened my children and me.”* (FGD, Female, 35)

Due to the worsening water situation, infants and mothers became weaker. As reported, many mothers became unable to produce milk in sufficient quantities for the babies, shortening the frequency and duration of breastfeeding, as shown in Figure 1. Stress was another outcome of the insecure water situation. The constant tension and feeling of scarcity of water made many of the mothers mentally sick. The frequency of breastfeeding was consequently reduced, and babies felt irritation. Mothers reported low blood pressure in such water situations in vulnerable areas. Mothers who often experienced extreme levels of water insecurity in their households reported increased dehydration and reduced breastmilk production as serious issues for which they had no immediate solution. These mothers expressed, with consternation, that constant stress greatly influenced the quantity of breastmilk.

*“Water increases our nervousness if not available in abundance, and often the same type of foods are cooked; mother is like land, and child is like a plant; with water, they grow, otherwise become dry. We can’t buy water; therefore, we’ve to wait in long queues, and it causes stress and weakness. In crop cultivation and harvesting seasons, we leave infants and young children at home for a very long time, which impacts feeding and care.”* (FGD, Female, 25)

*“I am weak, and extra energy is required to fetch water in heavy cans, especially during pregnancy and lactation. During summer [June, July, and August], breastfeeding is reduced because there is a lack of water in the house, and the water situation affects travel and care, which increases stress.”* (FGD, Female, 19)

*“Breastfeeding practice is disturbed when we are fetching water, the burden of domestic work, leaving babies at home, knowing very well that nobody will take proper care of my children, and this anxiety makes my breastmilk dry as there is no permanent solution of water. When I go somewhere, family members don’t take care of my children.”* (FGD, Female, 28)

*“Current sources of water are open to humans, children and donkeys drink from the same source. These animals urinate in the water and cause life-threatening health situations in the community. Also, worms are found in drinking water that feels nasty. Due to this, unclean water is drunk in little quantity, which produces a reduced quantity of breastmilk. As we have to work hard and walk far away, it also impacts our kidneys when thirst is not quenched.”* (FGD, Female, 35)

### 3.4. Water Insecurity Cause Food Insecurity

#### 3.4.1. Low Milk Production

Animals produce little milk when water is at risk. Their fodder is critically reduced. In severe water insecure areas, people had to quench the thirst of their livestock so that they could produce a sufficient quantity of milk from them. When there was a shortage of water, animals remained thirsty and produced a severely reduced quantity of milk. Cattles were moved to the other villages to be able to drink just one time a day. The situation became much more critical in months when water reservoirs were too close to drying up. One informant remarked:

*“When we drink a large quantity of water, it seems we look fresh and our breasts produce a large quantity of milk. The same case is with our animals when they do not drink the water they look skeleton, poor and they produce a small quantity of milk.”* (KII, Female, 53)

#### 3.4.2. Low Agricultural Production

Because of brackish water, low agricultural production that endangered nutrition was also reported. Rajanpur, the Western side of the district adjacent to the Suleiman mountain range, is considered to be highly water-insecure. Brackish or hard water is also not suitable for agricultural purposes. Therefore, food became expensive and hard to obtain for the inhabitants. They had to travel long distances to satisfy water and food requirements. Sometimes, when there was a lack of money, it became difficult to decide between water and food as both are equally compulsory. The crops needed plenty of water to grow; hence, the low amounts of available of water were unable to fulfill the requirements, and crops therefore suffered the most. One man stated:

*“When water is short we can’t grow our crops… In result, poverty and hunger increase, and our young males ultimately have to immigrate to earn some money as a survival strategy.”* (FGD, Male, 40)

### 3.5. Locals’ Understanding of Pathways and Possible Suggestions

Community members expressed implications of water insecurity through three major pathways (Figure 1). Based on this, they suggested that the government must take necessary and urgent steps for this disadvantaged community of the Punjab province of Pakistan. First, water in canals should be equally and justly provided throughout the year as it is available in the central region of the province. It is very necessary for sustainable agriculture, food security, and income-generating purposes, especially for small landholders and the Western inhabitants of the district who often become the victims of this water inequity. In addition, floodwater from the Suleiman Mountain range, which is always wasted, can be stored with the construction of a small dam. Second, a clean water supply system for drinking purposes should be installed near community residences as this will not only save younger children from infections but also women and girls from harassment, stigma, work burdens, and stress. In this way, mothers would be in a better position to take care of themselves as well as their children. Third, it could improve infants’ feeding and hygiene practices. 

## 4. Discussion

This study explored and analyzed the causes of the bad water situation and its effects on the health and nutrition of women and children in the Rajanpur district. Household water insecurity, as it influences hygiene, sanitation, domestic needs, feeding practices, care, educational attainment, economic opportunities, and overall social development, has become the central determinant of poor maternal and child nutritional status. The burdens of fetching water and managing difficult water situations affect households’ and mothers’ care capabilities and opportunities, and thus, lead to morbidity and malnutrition, devalued identity construction, time poverty, and sub-optimal care and feeding practices [10].

Good water governance has been described as a basic determinant of household water security [6,7,8,9,10]. It shows that at the household level, water insecurity has a political context that influences WASH, IYCF, food security, and maternal mental health along with gender inequities. Water insecurity has formed complex and syndemic relationships.

The findings showed that the canal water was available for only a few months of the year in the Rajanpur district; however, in North-Punjab, canals were supplied with water throughout the year. People in the district were deprived of a due water share and could not receive a just supply of water both for crop production and household consumption. Further, evidence authenticates that water theft, either in the form of Moga meddling or out-of-turn water taking, is dealt with by officials of the Punjab Irrigation Department, without involving the police or the judiciary. In addition, the most powerful landlords are given undue favors and unfair proportions of water by the Irrigation Department as the distribution of canal water to “big landlords” has never been fairly audited. Larger post-colonial water distribution policies that developed in colonial times seem to have influenced water injustice at micro-levels [28].

People from these areas showed helplessness as they only waited for the situation to improve before seriously undertaking the migration plan. When there was no water, people started to migrate. They became exhausted during long hours of travel, their immunity was lost, and they ultimately became exposed to diseases. Animals and livestock were taken to the other villages for quenching their thirst, which affected their milk production. Agriculture and livestock required sufficient amounts of water, which was not available there [29,30]. As a result, crop yields reduced [31,32,33] and animals’ fodder quantities became less [34], which caused serious implications for household food security.

River (Indus) water is widely used in the district. Evidence showed it was contaminated with coliform, fecal coliform, and E. coli [35,36], which was a major threat to public health. Khan et al. [37] found that coliform bacteria caused gastroenteritis, dysentery, diarrhea, and viral hepatitis. Many of the locals cleaned defecation wastes with either mud or a pebble, which posed a danger in terms of feeding.

Studies indicated that in most regions of Pakistan, the water supply system was outdated. Old and leaking pipes caused contaminated water supplies [38]. The political economy links water insecurity with weak infrastructure [39]. People had to pay an inflated amount for bottled water when infants became sick. Evidence shows that the purchase of water makes it difficult to decide between water and food [21].

In addition, the results indicated that there was a stigma related to water fetching. In some places, the better-off people made a mockery of the poor. They often forced the poor to engage in labor or received domestic labor from them in compensation for water. Poorer residents also sometimes had to listen to rude and abusive words. Girls were more frequently told that they could receive water free of charge if they were willing to engage in illicit sexual relationships. Girls felt danger, especially after sunset, when travelling to fetch water. Husbands often felt angry at wives if they were late in bringing water. Females, therefore, usually prefer fetching water in groups in daylight and tend to avoid encountering strangers. This situation also exposed young girls to stigmas, as well as physical and sexual violence [40]. Moreover, trivial fights grew into family scuffles and feuds. The main reason for the clashes was the women wanted to save their time, as they became tired of waiting in the queue for the water. Several interpersonal conflicts were reported [14,21] due to the problem with water, and many of the respondents felt angry [10,41]. The water crisis affects not only women but also everyone. However, women, being more susceptible, become the major victim of this structural violence [42].

Females fetched water during pregnancy, delivery, and lactation [21]. Mothers concluded that they had to travel, ignoring care and recovery, and breastfeeding was either stopped because of a lack of water or its frequency was reduced, and babies felt irritation. They also observed that anxiety made their breastmilk dry, and they experienced low blood pressure and nervousness. Further, fetching water during sickness impacted not only the mother’s well-being but also the infant’s health [43]. This reduced their daily caloric requirement in the dry season and increased fatigue and other physical illnesses [44]. Additionally, breastfeeding was difficult due to water fetching [18], and breastmilk production and breastfeeding frequency were consequently reduced [45,46,47]. Many mothers reportedly left their children at home while fetching water, which impacted care and proper feeding [48]. The study conducted by Aihara, Shrestha, and Sharma in Nepal corroborated these findings [13], as participants worried that they would not have enough water, so they consumed small amounts of water and faced a lack of hygiene and less time for childrearing.

In the end, it is clear from this empirical investigation that the water and nutrition nexus involves collaborations from several other sectors including agriculture, food, gender, public health, and political economy. A recent review study showed similar conclusions, namely that an interdisciplinary collaboration in the water–nutrition nexus is vital for ensuring equitable access to safe water and healthy foods [49].

By employing FGDs and KIIs (qualitative research methods), this current study contributed to the discussion on how a politics of neglect and regional disparities in terms of water culminated in experiences of gender inequities, food insecurity, and suboptimal feeding practices that ultimately aggravated maternal health and child nutrition in the district. The methodology showed some limitations, and the quality of data might have been impacted during the research process; for example, questionnaires were first asked in the local language and then translated again into the English language. Next, the selection of participants was based on purposive and snowball sampling that might restrict the proper representation of the local population. Finally, the researcher who belonged to that area could have shown some reflexivity during the data collection and analysis processes.

## 5. Conclusions

Along with insufficient water supplies, perceptions of mismanagement of water supply by government entities were highlighted in this article. Insecure and unsafe water at the community level in the Rajanpur district is a result of political indifference and regional inequities in Punjab province. Water from boreholes and canals is perceived to have poor taste and quality. People complained about the negative quality and quantity of water. Sanitation and hygiene are other factors connected to sufficient water quantity. Due to a lack of water availability, the practice of open defecation was prevalent. Water emerged as a cross-cutting theme that increased gender inequities through work burden, reduced maternal care, and child feeding. It is essential to highlight that the government should immediately take significant steps for this impoverished community as sustainable development projects have always been deprioritized because of regional and structural inequities in the Punjab province of Pakistan. This research illustrated the relationship between water insecurity, gender inequality, and malnutrition. Thus, it is water justice that is most fundamental because it has syndemic relationships with optimum WASH, IYCF practices, maternal mental health, food security, and subsequently poor breastfeeding behavior at the community level. Food production, gender equality, and a sustainable water situation would certainly safeguard a considerable reduction in malnutrition in these areas.

## Figures and Tables

**Figure 1 ijerph-18-12534-f001:**
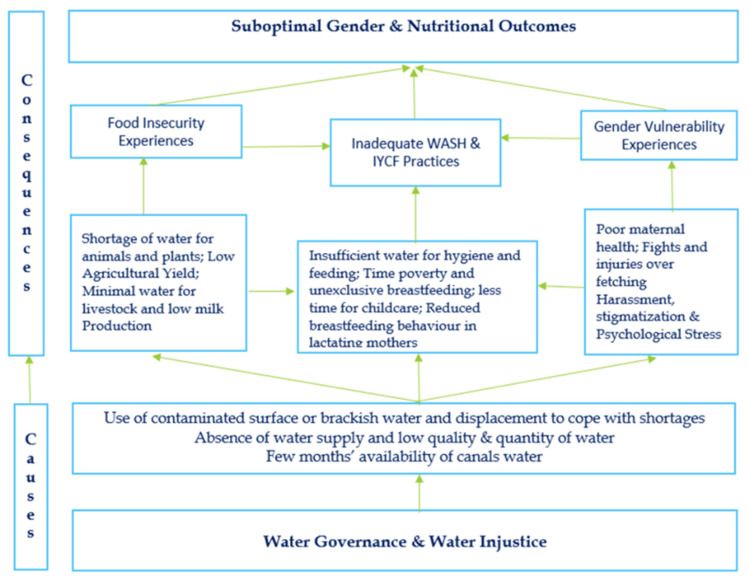
Locally described probable linkages and pathways showing causes of water insecurity experiences and their gender and nutritional consequences in the Rajanpur district of the South-Punjab in Pakistan.

**Table 1 ijerph-18-12534-t001:** Socio-demographic Characteristics of FGDs and KI Participants (*n* = 35).

Description of Participants	No of Participants (*n*)
FGD 1 (with Community Females)	10
FGD 2 (with Community Males)	10
FGD 3 (with Lady Health Workers)	10
KII (3 Females and 2 Males)	5
Socio-demographic Characteristics (*n* = 25)	Frequency (%)
Gender	
Female	23 (65)
Male	12 (35)
Literacy	
Illiterate	18 (51)
Primary to Middle	3 (9)
High	14 (40)
Occupation	
Agricultural Labor	14 (40)
Domestic Labor	8 (22)
Salaried	13 (37)

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
