# Peer review of "A Qualitative Exploration in Causes of Water Insecurity Experiences, and Gender and Nutritional Consequences in South-Punjab, Pakistan"

_ijerph, 2021, doi:10.3390/ijerph182312534_

Round 1
Reviewer 1 Report
Congratulations on presenting an excellent research paper on the topic of water justice. you could improve this paper by adding only one or two things. First, there should be very clear, what is the purpose of your study? And what are your research questions that you are wanting to understand? This should be presented in the methodology, and shown how this research answered this. Second, there should be a section on application - or suggestions - surely the participants have give you some ideas of what to do to improve the sitatuation. Add this to the results. Excellent papaer on a veruy important topic!
Author Response
Dear editor and reviewers
We are very thankful for your time, interest and valuable comments and suggestions you provided during the review process of our paper. These comments are highly useful for revising and improving the quality of our manuscript. We have carefully edited our paper and removed language issues. All the sentences have now been critically reviewed once again to improve the write-up.
Congratulations on presenting an excellent research paper on the topic of water justice. You could improve this paper by adding only one or two things. First, there should be very clear, what is the purpose of your study? And what are your research questions that you are wanting to understand? This should be presented in the methodology, and shown how this research answered this. Second, there should be a section on application - or suggestions - surely the participants have given you some ideas of what to do to improve the situation. Add this to the results. Excellent paper on a very important topic!
Response of first: Thank you so much for your comments. A very clear statement on purpose of the study and research question/objective has been incorporated and placed in methods section; and it clearly shows how our research has answered this. This research aims to explore whether higher prevalence of poor health and nutrition in a marginalized district of South-Punjab province (Rajanpur) of Pakistan is associated with water injustice.
Response of second. In the end of result section local suggestions and identification of pathways has been added (See heading 4.5. and figure 1 at the end of result section)
Reviewer 2 Report
Dear authors, here are a few suggestions for improving the manuscript:
- The first part of the methodology (study setting) would perhaps be better placed at the end of the introduction.
- The limitations that are placed in the methodology should be at the end of the article.
- It would be interesting to have a table relating the questions and categories obtained during the analysis.
- It would be easier to read if the data collection and data analysis sections were separated in the methodology.
- The bibliography is not adapted to the standards of the journal (name of the journal in italics, year of publication in bold, etc).
Author Response
Dear editor and reviewers
We are very thankful for your time, interest and valuable comments and suggestions you provided during the review process of our paper. These comments are highly useful for revising and improving the quality of our manuscript. We have carefully edited our paper and removed language issues. All the sentences have now been critically reviewed once again to improve the write-up.
Dear authors, here are a few suggestions for improving the manuscript:
- The first part of the methodology (study setting) would perhaps be better placed at the end of the introduction.
- The limitations that are placed in the methodology should be at the end of the article.
- It would be interesting to have a table relating the questions and categories obtained during the analysis.
- It would be easier to read if the data collection and data analysis sections were separated in the methodology.
- The bibliography is not adapted to the standards of the journal (name of the journal in italics, year of publication in bold, etc.).
Authors’ Responses: Thank you very much for all these valuable comments. Please see our response below:
- The first part of the methodology (study setting) has been placed at the end of the introduction.
- Limitations of study previously placed in the methodology section are now at the end of the discussion section of article.
- The categories obtained from questions during the analysis have been given at the end of result section (See Figure 1 at the end of result).
- Data collection and data analysis sections have been separated in the methodology section of revised paper. (See Methods section please)
- We revised the bibliography section and it is now adapted to the standards of the journal. The name of the journals is given in italics, and year of publication in now bold, and also other changes have been incorporated as required by the journal) (See references section please)
Reviewer 3 Report
Journal Article Review
Title: A Qualitative Exploration of the Causes of Water Insecurity Experiences, and Gender and Nutritional consequences in South-Punjab Pakistan
Authors: Farooq Ahmed, Muhammad Shahid, Yang Cao *, Sidra Zia, Madeeha
Qureshi, Saireen Fatima, Jing Guo
Thank you for the opportunity to review the above referenced manuscript.
As the title suggests, this manuscript provides a qualitative discussion of water insecurity in South-Punjab Pakistan as it relates to the impact on mothers’ physical and mental health and infants and young children’s feeding. The authors used focus group discussions and key
informants to describe local experiences of water insecurity issues.
Introduction
Sustainable Development Goal Six is mentioned with no point of reference. The authors might want to include reference to the United Nations Sustainable Goal Six (line 36) for clarity
purposes. The goal as set forth by the United Nations is to “ensure availability and sustainability…” The authors stated that the goal “guarantees” Goal Six which is slightly misleading.
The authors describe the worldwide problem of contaminated drinking water faced by 1.8 billion people. The authors further suggest that proper community development interventions are needed and that low and middle income countries the peripheries are deliberately ignored in terms of human development. The manuscript focuses on the deprived district of Southern
Punjab.
Several articles are cited which describe issues of stunting rates and height-for-age rates that are below normal and blamed on inadequate water quantity. Improper hygiene practices in women and their children are a source of stress for the mothers.
The authors state that brackish water used for drinking contains arsenic that turns breastmilk poisonous. (Line 61) This may be a generalization based on published literature and is not necessarily a fact for South-Punjab. The authors might want to rephrase this sentence something like “The literatures suggests that arsenic contamination of drinking water is a possibility that may result in contaminated breastmilk.”
Line 64 refers to water scarcity “…hit women and children on critical days.” The authors might want to define a critical day.
The authors state that while there is available literature on water scarcity there is limited information on the impact of water scarcity in the poorer southern communities of Pakistan. The aim then of the manuscript is to discuss the households’ experiences of poor water conditions in general and the impact of water fetching and borrowing on maternal stress as it relates to infant feeding. The authors aim to investigate household experiences regarding the relationship of water’s importance as it relates to optimal maternal-child health and nutrition.
Materials and Methods
The authors focused on Rajanpur, one of the least developed district of Punjab province with a high maternal-child prevalence as well as extreme poverty. Rajanpur presents a picture of Punjab with insufficient health and education facilities and an increased poverty rate. (Line 99)
The primary sources of drinking water include rain and canal water. Canal water is available only six months out of the year and underground water is brackish and undesirable by inhabitants of the area.
Three focus groups (community females, community males and Lady Health Workers) of ten persons and five key informants were interviewed for the manuscript. The key informants provided information and relevant data about the area, beliefs, practices and experiences of water insecurity and its causes and consequences.
After intensive interviews, transcripts of the interviews were translated into English. The authors list two limitations, first that the questionnaires were translated from the local language and second, participants were selected based on snowball sampling. The local researchers may have introduced sampling bias.
According to Table 1, 51% of the participants were illiterate. Informed consent was obtained by oral consent. The study did have IRB approval.
Results
Study participants to lower socioeconomic status and were employed as agricultural laborers or domestic servants.
Results of the questionnaires were broadly divided into three categories:
• Water governance and Coping Strategies
• Water borrowing and fetching cause gender vulnerability
• Water insecurity impacting food insecurity, suboptimal water, sanitation, and hygiene and infant and young children’s feeding
• Water insecurity as a cause of food insecurity
Discussion
The study analyzed the effects of poor water quality on the overall health and nutrition of women and children in Rajanpur.
Canal water as a source of water (inadequate quality) was only available a few months of the year in the Rajanpur district; however, in North-Punjab canals are supplied with water year-round. The study identified politics of neglect and regional disparities in terms of water availability resulted in gender inequities experiences, food insecurity ad suboptimal feeding practices that ultimately aggravated maternal health and nutrition in the district.
Conclusions
In the district of Rajanpur:
• Insufficient water supplies were observed
• There is a perception of mismanagement of water supplies by government entities
• Water from bore holes and canals is perceived to have poor taste and quality.
• Sanitation and hygiene are issues.
• Gender inequities exist
• Government intervention including significant steps including sustainable development
projects are needed to reduce the effects of water insecurity, gender inequality, and malnutrition. These actions are needed to reduce malnutrition in this community.
References cited in the manuscript are current and satisfactory.
Line 36 references Sustainable Development Goal Six. This may not be familiar to all readers You might want to clarify where the goals come from.
The manuscript includes many run-on sentences which could be divided into individual sentences which would make the manuscript more readable.
Author Response
Dear editor and reviewers
We are very thankful for your time, interest and valuable comments and suggestions you provided during the review process of our paper. These comments are highly useful for revising and improving the quality of our manuscript. We have carefully edited our paper and removed language issues. All the sentences have now been critically reviewed once again to improve the write-up.
Comment: Sustainable Development Goal Six is mentioned with no point of reference. The authors might want to include reference to the United Nations Sustainable Goal Six (line 36) for clarity purposes. The goal as set forth by the United Nations is to “ensure availability and sustainability…” The authors stated that the goal “guarantees” Goal Six which is slightly misleading.
Response: Thank you for your suggestion and comment. As you suggested reference has been provided and Word “guarantees” has been replaced with word “ensures” (see 1st paragraph of introduction please).
Comment. The authors state that brackish water used for drinking contains arsenic that turns breastmilk poisonous. (Line 61) This may be a generalization based on published literature and is not necessarily a fact for South-Punjab. The authors might want to rephrase this sentence something like “The literatures suggests that arsenic contamination of drinking water is a possibility that may result in contaminated breastmilk.”
Response: Thanks for your comment. Sentence has been rephrased as suggested: “Evidence suggests that arsenic contamination of drinking water is a possibility that may result in contaminated breastmilk.”
Comment. Line 64 refers to water scarcity “…hit women and children on critical days.” The authors might want to define a critical day.
Response: Thanks for highlighting this important point. We wanted to define severe water scarcity if it is more than one month in one year. In the revised manuscript we revised it as: “Evidence shows worldwide four billion (2/3rd) people face moderate to severe water scarcity at least one month in a year. Whereas in Pakistan, 120 million (>3/4th in the Indus basin) people face severe water scarcity for at least 4 to 6 months per year [16]. Severe water scarcity hit women and children on critical days of pregnancy and lactation.”
Comment: Line 36 references Sustainable Development Goal Six. This may not be familiar to all readers. You might want to clarify where the goals come from.
Response: Thanks for your valuable suggestion. We added and revised sentence as: “In order to tackle poverty, ill health, inequality and environmental degradation, Sustainable Development Goals (SDGs) were born in 2012 at the United Nations Conference on Sustainable Development in Brazil.”
The manuscript includes many run-on sentences which could be divided into individual sentences which would make the manuscript more readable.
Response: Thanks for your comment. We have revised and edited the manuscript once again. Wherever found the long and run-on sentences we have divided them into individual sentences in the revised draft.